# Synthesised Conductive/Magnetic Composite Particles for Magnetic Ablations of Tumours

**DOI:** 10.3390/mi13101605

**Published:** 2022-09-27

**Authors:** Chiang-Wen Lee, Ju-Fang Liu, Wen-Chun Wei, Ming-Hsien Chiang, Ting-Yuan Chen, Shu-Hsien Liao, Yao-Chang Chiang, Wen-Cheng Kuo, Kuen-Lin Chen, Kuo-Ti Peng, Yen-Bin Liu, Jen-Jie Chieh

**Affiliations:** 1Department of Nursing, Division of Basic Medical Sciences, Chronic Diseases and Health Promotion Research Center and Research Center for Chinese Herbal Medicine, Chang Gung University of Science and Technology, Puzi City 61363, Taiwan; 2Department of Orthopedic Surgery, Chang Gung Memorial Hospital, Puzi City 61363, Taiwan; 3Department of Safety Health and Environmental Engineering, Ming Chi University of Technology, New Taipei City 243, Taiwan; 4School of Oral Hygiene, College of Oral Medicine, Taipei Medical University, Taipei 11031, Taiwan; 5Department of Medical Research, China Medical University Hospital, China Medical University, Taichung 404, Taiwan; 6Institute of Electro-Optical Engineering, Gongguan Campus, National Taiwan Normal University, Taipei 106, Taiwan; 7Department of Anatomy and Cell Biology, College of Medicine, National Taiwan University, Taipei 106, Taiwan; 8Department of Mechanical and Automation Engineering, National Kaohsiung University of Science and Technology, Kaohsiung 81157, Taiwan; 9Department of Physics, National Chung Hsing University, Taichung 402202, Taiwan; 10Division of Cardiology, Department of Internal Medicine, National Taiwan University Hospital, Taipei 100229, Taiwan

**Keywords:** iron oxide, Ga, composite, tumour, magnetic ablation

## Abstract

Ablation is a clinical cancer treatment, but some demands are still unsatisfied, such as electromagnetic interferences amongst multiple ablation needles during large tumour treatments. This work proposes a physical synthesis for composite particles of biocompatible iron oxide particles and liquid metal gallium (Ga) with different alternative-current (AC)-magnetic-field-induced heat mechanisms of magnetic particle hyperthermia and superior resistance heat. By some imaging, X-ray diffraction, and vibrating sample magnetometer, utilised composite particles were clearly identified as the cluster of few iron oxides using the small weight ratio of high-viscosity liquid metal Ga as conjugation materials without surfactants for physical targeting of limited fluidity. Hence, well penetration inside the tissue and the promotion rate of heat generation to fit the ablation requirement of at least 60 °C in a few seconds are achieved. For the injection and the post-injection magnetic ablations, the volume variation ratios of mice dorsal tumours on Day 12 were expressed at around one without tumour growth. Its future powerful potentiality is expected through a percutaneous injection.

## 1. Introduction

Cancer treatment is performed on the basis of different types and stages of cancer, with complicated clinical requirements [1]. Cancer therapies can be categorised as surgery, radiation therapy, chemotherapy, and others. However, radiation therapy and chemotherapy have some side effects that damage healthy cells and tissues due to the non-precise treatment position [2,3,4,5,6]. On the contrary, tumour ablation causes cancer tissues to be exposed to high-temperature sources, such as ablation probes, to heat cancer cells with little damage to normal neighbour tissue in several minutes. The biological mechanism is to promote cell death through coagulation necrosis in different thermal doses of high temperature levels and heating time. With precise and mini-invasion, some electromagnetic methods had been proposed to generate heat for inoperable patients with other comorbidities [7,8,9,10,11,12]. Radiofrequency ablation (RFA) is the current primary electromagnetic-based ablative modality of releasing high-frequency current in abdomen lesions, such as hepatocellular carcinoma [7,8,9]. Microwave ablation (MWA), another alternative electromagnetic method, induces tumour destruction by agitating water molecules in cancerous tissues to produce friction heat [10]. Although these tumour ablations have been used in clinical practice, some limitations include tumour size, long ablation time, and similar uncontrollable biosafety risks of releasing high-frequency electromagnetic energy to biological tissue with radiation therapy and chemotherapy. Hence, injectable materials with the absorption of emitted electromagnetic for more precision ablation were recently emphasised. Furthermore, the multiple injections of ablation materials were much simpler and more workable for simultaneous absorption of electromagnetic energy than the multiple ablation probes of RFA or MWA. Contrary to the shallow depth limitation of light penetration for optical materials [11,12], magnetic materials were workable for deep lesions by magnetic fields. Due to the well-known biocompatible, iron oxide particles were popularly used to form different types of nanoparticles in cancer treatment or other biomedical applications, such as drug carriers [12,13], magnetic particle hyperthermia for lower heating temperature [14,15,16], magnetic ablations using magnetic nanoparticles [17,18,19], the biomedical detection or so-called magnetic immunoassay reduction of magnetic labelling [20,21], image contrast of magnetic resonance imaging [22], ultrasound-induced magnetic imaging [23], etc. These applications of the absorption of AC magnetic energy by magnetic nanoparticles contributed to the magnetization energy majorly and the heat energy slightly based on the in-phase and out-of-phase AC magnetic susceptibility of magnetic particles [24]. Hence, due to the non-efficiency of heat generation, magnetic particle hyperthermia, rather than magnetic ablations, was almost achieved and then used as the released energy of carried molecular or drugs [12,13]. Here, hyperthermia and the ablation of thermotherapy were separately defined as the lower and higher temperatures of the heated tissue between 40–48 °C and 60–90 °C [25]. Actually, the higher concentrations of magnetic nanoparticles or the higher frequencies or amplitudes of AC magnetic fields theoretically promoted the heat generation rate due to the so-called Neel or Brownian relaxation loss as the induced friction heat from out-of-phase AC susceptibility. However, the biosafety considerations made both mentioned promotion parameters limited. For example, the biosafety criteria of AC magnetic fields were the frequency of the generated magnetic field (f) should be less than 200.0 kHz, and the product of the intensity of the field (H) and f should be less than 5.0 × 10^9^ A/m/s [26,27,28]. Some works with the achievement of magnetic ablations using magnetic nanoparticles occurred with no consideration of these mentioned limitations or low-water contained materials, such as bones [17,18,19].

The promotions of biomedical performance for iron oxide particles focused on the method using different solvent materials or liquid-based composite materials, instead of the over-large specifics with biosafety risks in conventional water-based solutions. For example, the so-called ferrogels have been developed for giant magneto-impedance sensors by introducing magnetite particles of colloidal size into chemically crosslinked poly hydrogels [29,30]. In addition, liquid metals have been widely used in many industries due to several superior properties, such as low melting points and high electrical and thermal conductivities. Especially for gallium (Ga), biocompatibility was additionally important for further tumour applications, including image contrast of ^67^Ga-citrate [31,32]. Similarly, Ga particles, as the carriers of therapeutic materials, can be metabolised through the excretory system in vivo in the first 30 days, indicating their biodegradability in mammals [33]. Different from the utility of therapeutic materials, the magnetic ablation using liquid metal Ga for different body parts of rats showed several superior performances, including the induced heat mechanism of AC magnetic fields and biodegradability [24]. However, the combination of liquid metal Ga and iron oxides is seldom studied, and studies on thermal ablation therapy are unavailable [34]. Therefore, this work developed biosafety, noncontact and efficient magnetic ablation techniques using biocompatible composite materials of liquid metal Ga and iron oxide particles under AC magnetic fields of biosafety specifics. In addition to the material characterisation of particle geometries, DC magnetics and AC-magnetic-field-induced heat of in vitro and in vivo tests, the biological studies of post-treatment include time-variation tumour sizes, the whole lying body and tumour tissue stain.

## 2. Materials and Methods

### 2.1. Synthesis of Composite Particles

Micrometre particles of liquid metal Ga and iron oxide composite were synthesised from liquid metal Ga (5 N, Kunshan Zhangpu Town Weiju Trading Firm, Kunshan, China) and iron oxide particles (325 Mesh Powder Fe_3_O_4_, Alfa Aesar, MA, USA). Initially, 300 mL of liquid metal Ga at 80 °C and 0.5 g of iron oxide particles were mixed on a 6 mm plastic plate (SADZK0007, Kartell, Italy) by magnetic stirring (MS-500, Chrom Tech, MN, USA). The mixture was stirred clockwise and counterclockwise at 2500 rpm by using an iron pin of 0.8 and 15 mm in diameter and length, respectively, as a stir bar instead of the original stirring bar to attractively move iron oxide particles into liquid metal Ga of high surface tension for strong physic conjugation for 5 min. Consequently, by a stainless mesh (120 mesh, Sinshe Metal, Taipei, Taiwan) with a filter hole diameter of approximately 120 μm, the mixtures were divided into two parts, i.e., the part with a smaller diameter than 120 μm was utilised for composite particles in all subsequent tests, and another part with larger-size ones than mesh holes for excluded composite particles in vitro tests of AC-magnetic-field-induced heat generation. The utilised composite particles of iron oxide particles and liquid metal Ga were obtained at approximately 0.4 g per batch.

### 2.2. Particle Geometry

A scanning electron microscope (SEM) (S-3000N, Hitachi, Tokyo, Japan), which was used to study the particle morphology, produced morphology images based on the interactions of electron beams with atoms at various depths within the sample. To study the component materials of liquid metals, iron oxides and composite materials of synthesised particles, these specimens were prepared by washing with deionised water for 5 min and then drying in an incubator. Finally, the specimens were observed under SEM for the analysis of size distribution and morphology.

### 2.3. Magnetics in Particle Scales

The magnetic force microscopy (MFM)/atomic force microscope (AFM)/scanning tunnelling microscope (STM) (SPA-400, Seiko Instruments, Chiba, Japan) of scanning probe microscope system was used to observe the distribution of composited metal and magnetic materials, and the features of magnetic domains and domain boundaries in the samples were elucidated [35]. Tip–sample magnetic interactions can be detected and used to generate the magnetic structure of the sample surface sample. The tip scans the surface within the close range of 10 nm to obtain the sample structure. It detects not only magnetic forces but also atomic and electrostatic forces of samples for MFM and morphology images separately. Pure iron oxide particles and utilised composite ones were washed with deionised water for 5 min and dried in an incubator to prepare the sample particles. The samples were dried and then subjected to analyse profiles and magnetic susceptibilities of individual particles through morphology images and MFM images.

### 2.4. X-ray Diffraction (XRD) Analysis

XRD (D2 PHASER, Bruker Co. Ltd., Billerica, MA, USA) was used to analyse the composition materials. In addition, the crystallite size (L) was analysed by K × λ/(B × cosθ) of the well-known Scherrer equation, where the X-ray wavelength (λ) and shape factor (K) are 0.15418 and 0.89, respectively, indicating the peak position (θ) and full width at half maximum (B) from the measured spectrum [36].

### 2.5. Magnetic Characteristics

A vibrating sample magnetometer (VSM) instrument was used to study the DC magnetisation characteristics of magnetic materials by a time-varying magnetic field [37]. The magnetic types can be defined by the so-called hysteresis curve including saturation, retentivity, coercivity, permeability and hysteresis loop. In this study, synthesised composite particles and pure iron oxide powders were examined through the VSM (Model 4500, EG&G, Hudson, MA, USA) at least two times to confirm whether the reliability measurement of one sample is sufficient and to determine their similarities and differences. The sample volume was 20 μL.

### 2.6. In Vitro Tests of AC-Magnetic-Field-Induced Heat Generation

The test materials were pure iron oxide powders, liquid metal Ga and the synthesised composite particles in the smaller and larger sizes groups than 120 μm in diameters. The sample of 20 μL was placed in a microcentrifuge tube (Microtube, Deltalab, Barcelona, Catalonia, Spain). The AC magnetic fields were low and high at 12,971.13 A/m at 52 kHz and 45,995.78 A/m at 43 kHz. The platform of the AC magnetic field generator was used to modify the capacity sub-unit based on the high-frequency induction heating machine (SP25A, Shenzhen Shuangping Power Supply Technologies Co. Ltd., Shenzhen, Guangdong, China). The temperature during magnetic ablation was measured through a nonmetallic fibre optic thermometer (OTG-280, OPSens Inc., Québec City, Quebec, Canada). The AC magnetic field generator was popular in industries for heating or cutting metals based on the resistance heat of eddy current flow [38].

### 2.7. Cell Line

Mimic human breast cancer cells (4T1) were purchased from ATCC and cultured in RPMI1640 medium supplemented with 10% foetal bovine serum and 100 units/mL of antibiotic–antimycotic. The cells were cultured in T25 flasks and incubated in an atmosphere of 5% CO_2_ at 37 °C.

### 2.8. In Vivo Tumour Tests

All mice were purchased from Chimera Bioscience (Taipei City, Taiwan). The subjects of the animal experiment were four-week-old BALB/c mice. All mice were stored in a specific pathogen-free facility. All experiments were conducted in accordance with the animal care guidelines of the Guide for the Care and Use of Laboratory Animals (National Institutes of Health Publication No.86-23, revised in 1985) and were approved by the Animal Care and Use Committee of Chang Gung University of Science and Technology.

Cancer cells were collected and diluted to 1 × 10^6^ in 100 μL. 4T1 cells were mixed with medium and Matrigel (ratio 1:1) and injected into two dorsal sides per mouse for two tumours. A total of 18 tumour growths in nine mice ranged from 70 mm^3^ to 140 mm^3^; all mice were divided into the experiment and control groups. In the control group, two dorsal tumours of one mouse were untreated for no possible effects on the mouse biology. In the experiment group, only one dorsal tumour of any mouse was treated, but its other dorsal tumour was not injected regardless of magnetic ablation. Here, the treatments on the eight tumours were injected with liquid metal Ga (two mice) and injected with the utilised composite particles (two mice); then, magnetic ablations were performed. Another four dorsal tumours on four mice (injected with iron oxide particles and liquid metal Ga) had the same conditions as the two dorsal tumours of the control group. In addition, four dorsal tumours on the last four mice (magnetic ablation after the injection) are considered the control group due to the verified biosafety of AC magnetic fields without any injection [24]. Here, the injection materials were pure iron oxide particles or composite particles of 100 μL, and the applied high field was only 45,995.78 A/m at 43 kHz by the same generator as the in vitro tests. In magnetic ablations after injection in the experiment group, the tumour temperature during magnetic ablation was measured through the same nonmetallic fibre optic thermometer with in vitro tests.

All mice were arranged for the measurement of tumour sizes as well as photos of lying whole body on Day 1 (i.e., the first day of in vivo tests, 4, 8, and 12). The execution days of X-ray images were similar to the photos, except for Day 12, because the X-ray instrument was unexpectedly repaired. Here, the tumour volume was based on the formula (V=a×b×c×π∕6), where *a*, *b*, and *c* were the geometric parameters of a tumour in length, width and thickness, respectively [39]. Hence, 10 tumours were without injection (regardless of applying AC magnetic fields or not) in the control and experiment groups for the reliable baseline of tumour growth (Table 1).

### 2.9. X-ray Imaging

Penetrable high-energy electromagnetic radiation has wavelengths ranging from 10 picometres to 10 nanometres, corresponding to frequencies within the range of 30 PHz to 30 EHz. In radiography, X-rays (CMP200, CPI, USA) are generated by an X-ray generator and projected toward an organism or object, which absorbs the radiation on the basis of its density and structural composition. Then, the X-rays that pass through the object are captured behind the object by a detector. In this study, mice were imaged by X-ray before and after injection with the synthesised composite particles or liquid gallium Ga to observe differences.

### 2.10. Hematoxylin and Eosin Staining

Tissue sections (8 μm in thickness) were prepared from paraffin-embedded tissues, and then deparaffinised in xylene, rehydrated in a graded alcohol series and washed in deionised water. The sections were stained with hematoxylin and eosin (H&E) to evaluate tissue morphology and quantify the damage.

### 2.11. Statistical Analysis

Values were reported as mean ± standard deviation of the mean. Statistical analysis between two samples was performed using Student’s *t* test. The statistical comparisons of more than two groups were performed using one-way ANOVA with Fisher–LSD posthoc test. In all cases, *p* < 0.05 was considered significant.

## 3. Results

### 3.1. Geometries and Composition Studies

The comparisons of particle sizes and morphologies were analysed between the composition materials of iron oxides and Ga as well as the synthesised particles (Figure 1A). The SEM images of a Ga drop and the iron-oxide-based particles (representative of pure iron oxide powders and the utilised composite particles) showed the smooth and streamlined profile drop in the oval long and short axes of 100 and 80 μm, rough profile spots in 10–40 μm and smooth profile spots or clusters in 20–40 μm. The Ga drop was relatively different from the iron-oxide-based particles including pure iron oxide powders and the synthesised composite particles. MFM and morphology images further distinguish the two iron-oxide-based particles (Figure 1B). The upper images had a larger magnification of 10 μm scale, and the lower images had a lower magnification of 5 μm scale. For the pure iron oxide particles, the same profiles of bar-shape spots in scales of 0.5–1 μm were shown between yellow spots in morphology images and black spots in MFM images. However, a large difference between these images was observed in the utilised composite particles. The former showed apparently larger yellow clusters in scales of 1–2 μm than the iron oxide particles, but the latter exhibited several high-density distributions of ultra-small black dots for constructing the same shapes of yellow clusters in morphology images (the links using green dash lines).

The XRD spectrum of iron oxide particles expressed the same numbers and diffraction angles of major peaks with the standard XRD pattern of Fe_3_O_4_ (JCPDS card no. 19-0629) in the inset (Figure 1C). That of the composite particles was similar but with weak peak intensities. However, that of the liquid metal Ga cannot illustrate the peak decrease in the composite particles. Hence, the compositions of the composite particles were almost contributed from iron oxide particles. Based on the well-known Scherrer equation, the analysed size distributions of iron oxide particles and utilised composite particles were 25.8–46.2 nm and 25.6–79.9 nm, respectively.

### 3.2. Magnetic Studies of DC Magnetisation and AC Magnetic Field Induction Heat

To clarify the magnetic performance of the two iron-oxide-based particles, the golden standard VSM was used to evaluate the magnetisation (M) variation with applied magnetic fields (H), so-called hysteresis or M-H curves. The hysteresis curves of pure iron oxide particles and the utilised composite particles demonstrated larger and lower saturation magnetisation at approximately 57.6 ± 0.7 and 66.6 ± 0.8 emu/g around 3100 Oe of similar hysteresis or M-H curves (Figure 2A). In addition, the two hysteresis curves commonly expressed the ultra-small hysteresis loss and the rapid increase rate of M with H before H achieved ±3100 Oe.

Subsequently, the AC magnetic field induction heats of pure Ga, pure iron oxide particles, and composite particles were discussed in low and high AC magnetic fields of 12,971.13 A/m at 52 kHz and 45,995.78 A/m at 43 kHz (Figure 2B). In addition, that of the excluded composite particles was discussed, considering the effect of particle sizes. The most rapid temperature increase rates of 21.4 ± 3.2 °C/s and 2.6 ± 0.4 °C/s in high and low fields were commonly exhibited by pure Ga (labelled with cross markers). Moreover, for synthesised composite particles in the high and low fields, the solid lines for the utilised composite particles were 13.5 ± 2.0 °C/s and 0.8 ± 0.1 °C/s, but the dash lines for the excluded ones were 9.2 ± 1.5 °C/s and 0.4 ± 0.1 °C/s. Especially, for the iron oxide particles (labelled with square markers), the temperature increase rates were 10.4 ± 1.6 °C/s and 0.6 ± 0.1 °C/s in high and low fields.

### 3.3. Intra- and Post-Treatment

The tumour ablation was studied for only the liquid metal Ga and the composite particles because the pure iron oxide particles of no fluidity were not injected into the tumour. Here, the injection volume into the dorsal tumours was 100 μL. For the temperature comparison between the injected pure liquid metal Ga and the injected utilised composite particles, the initial temperature rising rate was the same in the first 120-S duration, but stable at different levels, for example, 75.7 ± 9.4 °C and 55.5 ± 7.2 °C for the tests of liquid metal Ga and the utilised composite particles (Figure 3A).

All 18 tumours of nine mice initially ranged from 70–140 mm^3^ on Day 1 (Table 1), and then the treatment effect was evaluated by the volume variation ratio of the average tumour volume on Day 12 over that on Day 1. For the non-injected in the black dash line (Figure 3B), the tumours continuously and apparently grew until Day 12, causing the volume variation ratio to become 5.7. For those injected with liquid metal Ga and utilised composite particles, their tumour growth curves of red and green dash lines were lower than that of the black dash line without injection, and their volume ratios were 4.6 and 1.9, slightly larger than 5.7 of that without injection. Furthermore, for the magnetic ablation after injection of the two materials, the tumour growth curves of red and green solid lines similarly oscillated with time since Day 1, and the volume variation ratios were 1.2 and 0.6 (Figure 3B).

The acquisition photos and X-ray images of mice after two post-injection AC magnetic fields are shown in Figure 3C. Especially, these enlarged images of dorsal tumours were shown as the inset over these images of their whole body for unequivocal visualization. Here, red dash lines and arrows were marked for the dorsal tumours with injection, i.e., the right shows the upper mouse (Figure 3C) injected with liquid metal Ga and the left shows the upper mouse (Figure 3C) injected with composite particles. In addition, yellow dash lines were marked for the dorsal tumours without injection. On Day 1, the treated tumours of liquid metal Ga or utilised composite particles were clearly observable in the photos due to their darker colour than their neighbours’ skins as well as in X-ray images due to the black spots or contours surrounding the injected tumour. On Day 4, the X-ray images show that the tumours and the black outer contours or inner spots were still grey and then disappeared. However, for the mice injected with liquid metal Ga, several white spots, marked with some red arrows, appeared on half the back body; two tumours also became visibly larger than they were on Day 1. On the contrary, the mice exhibited the same tumour growth as the one dorsal tumour without injection of the composite particles. On Day 8, the tumours without any injections, marked with yellow dash lines, became seriously larger than they were on Day 1, but the tumours maintained the same size in the photos and X-ray images. The differences in the complete spot shapes in MFM images could be explained as fluid-induced artefacts.

### 3.4. Stained Tumour Tissue

After packaging by slides/cover glasses, the H.E. stain slices of tumours with four treatments were observed at 10×, 40×, and 100× magnification (Figure 4). Firstly, when injected with liquid metal Ga, the black spots almost remained with large tissue spaces outside the tissue similar to microvessels. Secondly, the injected utilised composite particles were distributed between tumour cells and the extracellular matrix. These results showed the distribution of injected materials, rather than the damage-induced variation of tissue morphology. Thirdly, the magnetic ablation using liquid metal Ga resulted in the lost nuclei of different morphology in the boundary tissue next to the microvessel spaces with liquid metal Ga, i.e., the density of purple spots in the microscopy image of 100× magnification was much less than those of the two treatments of injection. Fourthly, the magnetic ablation using the composite particles exhibited seldom or no purple spots, i.e., the tissue was seriously damaged due to excessive nucleus loss.

## 4. Discussion

The XRD pattern of utilised composite particles reasonably agreed with the XRD peaks of the two compositions due to the composition of biocompatible iron oxide particles and liquid metal Ga, only the XRD spectra of iron oxide particles and without those of non-crystalline liquid metal Ga (Figure 1C). Furthermore, the average grain sizes by the Scherrer equation were from 25.8 nm to 46.2 nm for iron oxide particles, but from 25.6 nm to 79.9 nm for utilised composite particles. The finding indicated that the utilised composite particles may be composed of two large iron oxide particles with few liquid metal Ga due to the size ratio of approximately two times between the large utilise composite particles and the large iron oxide particles. Although the sizes of the utilised composite particles and iron oxide particles in MFM and morphology images (i.e., 0.5–1 μm and 1–2 μm for iron oxide particles and utilised composite particles, respectively, as shown in Figure 1B), were apparently larger than the analysed sizes from XRD spectra due to the aggregation during sample preparation, a similar size ratio of two for the utilised composite particles over iron oxide particles was maintained. Moreover, for iron oxide particles, the complete dark spots in MFM images are the same as the yellow ones in the morphology image. However, for the utilised composite particles, several small brown spots in MFM images, rather than complete spots as iron oxide particles, constructed the same shape with one large complete black spot in the morphology image. The differences in the complete spot shapes in MFM images could be explained as fluid-induced artefacts, such as the nonstable touch of MFM probes on utilised composite particles of non-rigidity due to the softness of the liquid metal Ga [40]. Similar to the MFM and morphology images, the sizes of the utilised composite particles and iron oxide particles in SEM images (i.e., 10–40 μm and 20–40 μm for iron oxide particles and utilised composite particles, respectively, as shown in Figure 1A) were apparently larger than the analysed sizes from XRD spectra based on the similar aggregation reasons of sample preparation. However, the similar size ratio of two for the utilised composite particles over iron oxide particles was maintained. Moreover, the spot boundaries and shapes in SEM images, (i.e., the smooth profile spots or clusters for utilised composite particles and rough profile spots for iron oxide particles) were different. Hence, these particle analyses supported clear images of utilised composite particles from the composition materials and the wide-size distribution per bath from several nanometres to micrometres.

The hysteresis curve (Figure 2A) indicated that the utilised composite particles were actually iron oxide particles with conjugation materials of few liquid metal Ga because their hysteresis curves were almost the same by the tiny difference of saturation magnetisation. Hence, the utilised composite particles possessed the same superparamagnetic properties as iron oxide particles [41]. Moreover, the ratio of the liquid metal Ga of non-magnetics in utilised composite particles could be reliably assumed as the weight percentage of 13.4% according to the average saturation magnetisation difference between utilised composite particles and iron oxide particles over the average saturation magnetisation of iron oxide particles, much larger than the measurement uncertainty of 1.1% (Table 1). The similar DC magnetic properties indicated that the image contrast functions of iron oxides in some biomedical imaging also worked regardless of the transportation in circulations.

Subsequently, the AC magnetic-induced heat of the composition iron oxide particles and liquid metal Ga belonged to different induced heat mechanisms of AC magnetic fields, i.e., the well-known magnetic particle hyperthermia and the giant resistance heat of the conductive materials, especially the superficial flow of eddy current [24,38]. The superior heat-generation performance of the latter mechanism has been verified to explain the best temperature raise rates with time [24]. For example, for high and low fields of in vitro tests, the temperature increase rates of liquid metal Ga were 21.4 ± 3.2 °C/s and 2.6 ± 0.4 °C/s, much higher than those of iron oxide particles around 10.4 ± 1.6 °C/s and 0.6 ± 0.1 °C/s. In comparison with iron oxide particles, the temperature increase rates of the composite particles were promoted to 13.5 ± 2.0 °C/s and 0.8 ± 0.1 °C/s for high and low fields, respectively, especially the increment of 29.8% at the high fields due to the incorporated liquid metal Ga of 13.4%. Hence, only high fields were applied to the following in vivo tests.

In addition, the utilised composite particles with smaller particles had better temperature rates in high and low fields than 9.2 ± 1.5 °C/s and 0.4 ± 0.1 °C/s of the excluded ones (Figure 2B). This finding is due to the superior field attraction that resulted from iron oxides, rather than the conductive liquid metal Ga. Hence, the low field was almost attracted by iron oxide particles for the well-known magnetic particle hyperthermia [14,15]. Here, the optimal heat generation of Neel or Brownian relaxation losses was exhibited by magnetic particles in tens of nanometres [42]. Consequently, larger sizes in several micrometres of the excluded composite particles were far away from large-size particles of the optimal heat generation. Moreover, the major part of the high field should be absorbed by conductive material for the giant resistance heat of superficial eddy current flow as the major contribution to the temperature increase rate after the minor part of the high field was sacrificed by the absorption of iron oxide particles for driving the direction of the magnetic field to liquid metal Ga. The excluded composite particles with more iron oxide particles obtained a higher ratio of magnetic field sacrifice in non-efficient magnetic particle hyperthermia [14,15] or magnetisation than the superior heat generation mechanism of the resistance loss in another composition of liquid metal Ga.

Actually, iron oxide particles without surfactant coating were unsuitable to be suspended in a water-based solution for in vivo injection. However, some liquid metal Ga, instead of surfactant coating and solvent, obtained movable iron oxide particles in addition to the heat-generation promotion in the utilised composite particles [14,15,20]. Furthermore, the limited fluidity of the utilised composite particles was reserved inside the tumours for the following magnetic ablation for several minutes. Hence, the physical quasi-targeting phenomenon after the injection into the object lesions, as the percutaneous implantation, was opposite to most biomedical treatments based on magnetic particles after circulation in blood vessels, such as drug delivery and biological targeting onto specific molecules. In other words, the transportation-related specifics, such as particle aggregation or suspension in water solvents and the uniformity of particle sizes, were not seriously required [22,43].

The temperature increase rates of utilised composite particles over the best ones of liquid metal Ga were 63.1% and 60.4% for in vitro and in vivo tests (Table 1). Although the consistent ratio of around 61.8% did not seem high, the tumour temperature after 10 min achieved 55.5 ± 7.2 °C close to the criterion temperature of 60 °C of the ablation level, where the tissue could be damaged in a few seconds. Here, the ablation time of 10 min was selected due to the common clinical ablation time using radiofrequency and microwave ablations for the tumour, viewed as the sphere in diameter of 3 cm in humans [44].

Different from the focus of the pre-/intra-treatment studies on composition/utilised materials, the treatment effects were biologically evaluated macroscopically by the variation of tumour sizes with the post-treatment time (Figure 3B), X-ray images and photos of the whole lying body at post-treatment days (Figure 3C), as well as microscopically by observing the stained tumour tissue on Day 12 (Figure 4). Due to high viscosity and surface tension [45,46], the liquid metal Ga, as drops in tens of micrometres, was distributed in tissue space similar to microvessels, rather than inside the tissue (Figure 4). On the contrary, the composite particles could be viewed as one cluster of a few iron oxide particles by the conjugation materials of little liquid metal Ga, rather than the biomaterials, such as antibodies and antigens (Figure 4) [47]. Hence, the composite particles still penetrated into the tissue effectively as individual iron oxide particles and then contributed to the so-called embolism in scales of nano-/micrometres to block the nutrition support to the tumour cells. Hence, only the injection of utilised composite particles had an acceptable volume variation ratio of approximately 1.9, i.e., 33% of 5.7 for no injection of any materials (Figure 3B). For those injected with liquid metal Ga, the nutrition transportation from microvessels and tissue space was similarly blocked due to the distribution of large Ga drops in scales of tens of micrometres. Consequently, the volume variation ratio of approximately 4.6 was 80% of 5.7 for no injection of any materials (Figure 3B). The relatively small embolism effect may result from the flow away from the tumour lesion due to the larger transportation ability of microvessels than tissue. The X-rays image on Day 4 verified that for a similar treatment of magnetic ablation after the injection, several liquid metal Ga were found in the back half of the whole body of red-arrow markers (Figure 3C). Moreover, the limited distribution of liquid metal in the large tissue space affected the magnetic ablation negatively when the composite particles were used, i.e., the volume variation ratio of 1.2 for the former was still two times 0.6 for the latter. However, the tumour temperature of in vivo tests for magnetic ablation using liquid metal Ga was apparently higher at 75.7 ± 9.4 °C than the utilised composite particles at 55.5 ± 7.2 °C (Figure 3A).

## 5. Conclusions

The utilised composite particles had a wide distribution of particle sizes in scales of tens of nanometres to micrometres due to high-viscosity liquid metal Ga and no surfactant coating. Small iron oxide particles of one composition material effectively penetrated into the tissue for the so-called embolism in scales of nano-/micrometres and the wide heater distribution for the heat conduction of the magnetic ablation induced heat. Liquid metal Ga of another composition material in as few as the weight percentage of 13.4% promotes the worse AC-magnetic-field-induced mechanism of magnetic particle hyperthermia from iron oxide particles in the increment of 29.8% at the high fields to fit the ablation requirement of 60 °C in several seconds. Eventually, the biological treatment effect verified the superior volume variation ratios of the injection and magnetic ablation using utilised composite particles to liquid metal Ga with a superior rate of temperature increase in vivo and in vitro tests.

## Figures and Tables

**Figure 1 micromachines-13-01605-f001:**
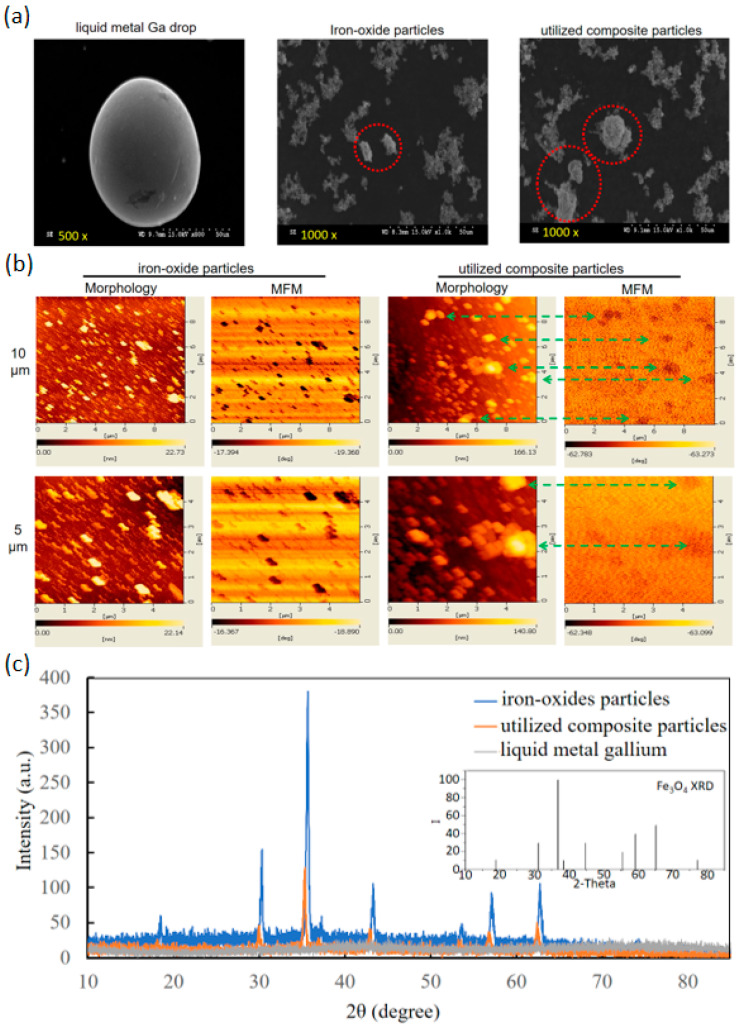
Characterisation of component materials and utilised composite particles. (**a**) SEM images of liquid metal Ga, iron oxide particles and utilised composite particles. Red circles were the representatives. (**b**) Morphology and MFM images of iron oxide particles and composite particles. The arrows linked the same particles in these two images. (**c**) XRD analysis of liquid metal Ga, iron oxide particles, and composite particles.

**Figure 2 micromachines-13-01605-f002:**
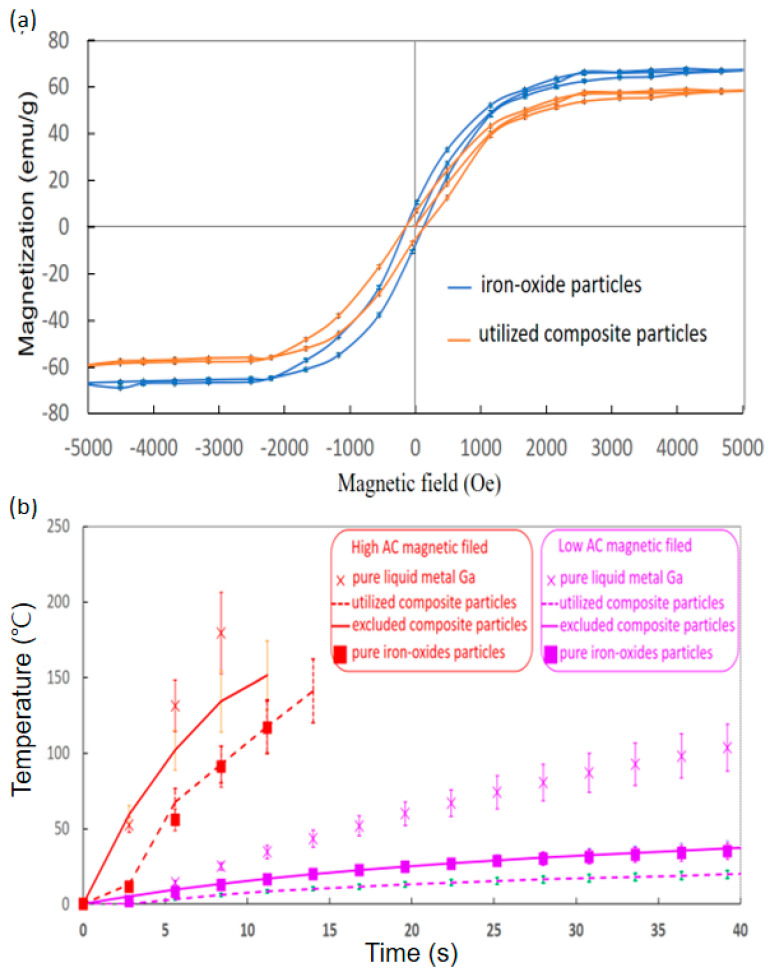
Characteristics of DC magnetisation and in vitro AC magnetic induction heat. (**a**) Hysteresis of iron oxide particles and utilised composite particles in 20 μL. (**b**) Temperature for liquid metal Ga and utilised and excluded composite particles in 100 μL under high and low magnetic fields (12,971.13 A/m at 52 kHz and 45,995.78 A/m at 43 kHz).

**Figure 3 micromachines-13-01605-f003:**
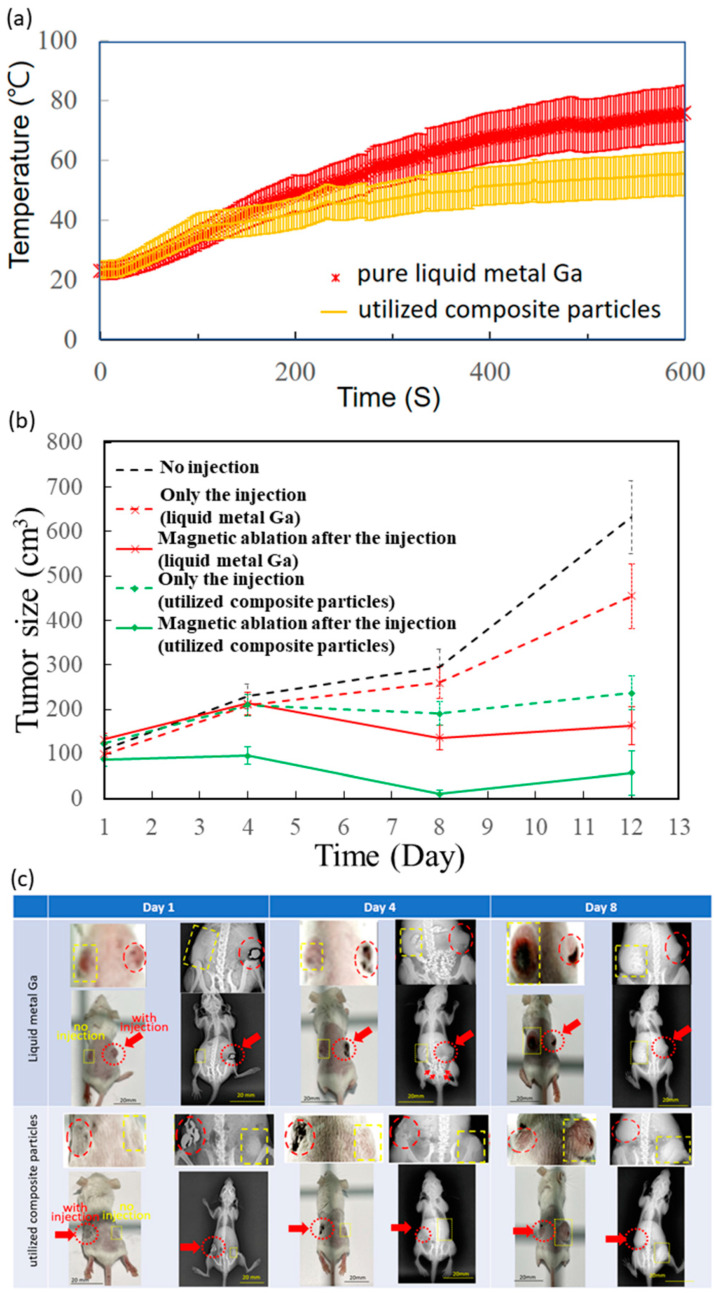
Intra- and post-treatments of in vivo tests. (**a**) Temperature variation with intra- magnetic-ablation time of tumour injected by liquid metal Ga and composite particles. (**b**) Tumour size variation with post-treatment time. (**c**) Photos and X-ray images of whole mice bodies in two magnetic ablations. Yellow dash lines were for no injections, and red dash lines and arrows were for injections.Four treatments were only the injection of liquid metal Ga and utilised composite particles as well as magnetic ablation after the injection of liquid metal Ga and utilised composite particles.

**Figure 4 micromachines-13-01605-f004:**
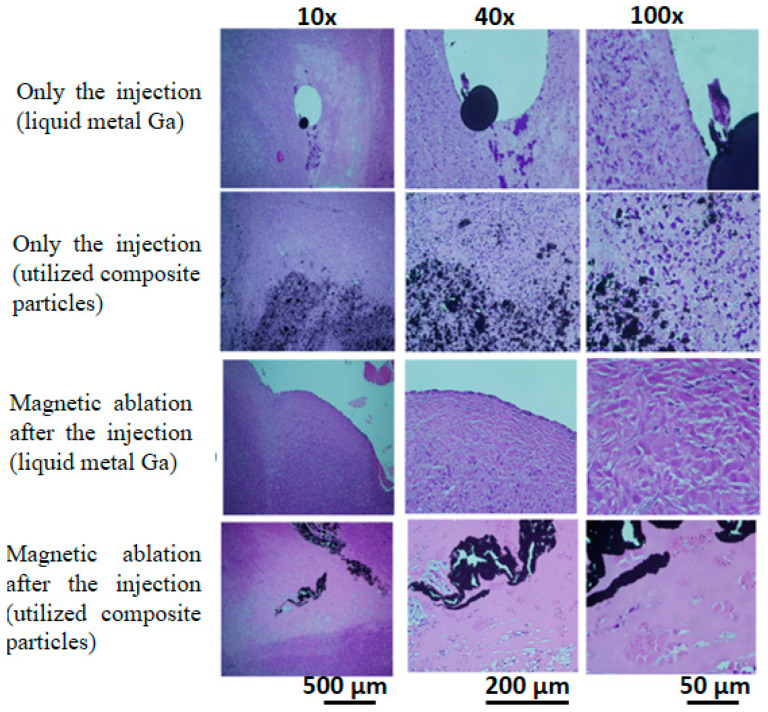
Stained tumour tissue using H&E stain for four treatments. Four treatments included the injection of liquid metal Ga and utilised composite particles as well as magnetic ablation after the injection of liquid metal Ga and utilised composite particles.

**Table 1 micromachines-13-01605-t001:** Different characterisations of composition/utilised materials in pre–/intra–/post–treatment.

Properties-Sample Sizes (Test Conditions)				Utilized Composite Particles	Iron–Oxide Particles	Liquid Metal Ga
**Geometry**	**individual particles (by SEM)**			**smooth-profile spots or clusters in 20–40 μm**	Rough-profile spots in 10–40 μm	the drop of smooth and streamlined profile in 80–100 μm
	calculated particles (by XRD)			range between 25.6–79.9 nm	range between 25.8–46.2 nm	unavailability
Magnetics	individual particles (by MFM & Morphology)			larger clusters in scales of 1–2 μm	bar-shape spots in scales of 0.5–1 μm	×
	Sample amount (by VSM)			57.6 ± 0.7 emu/g (saturation magnetization)	66.6 ± 0.8 emu/g (saturation magnetization)	×
AC-magnetic-field-induced heat generation	In vitro tests (high and low fields)			↑ 13.5 ± 2.0 °C/s and ↑ 0.8 ± 0.1 °C/s	↑ 10.4 ± 1.6 °C/s and ↑ 0.6 ± 0.1 °C/s	↑ 21.4 ± 3.2 °C/s and ↑ 2.6 ± 0.4 °C/s
	In vivo tumor Tests	Exp. Group	Magnetic ablation @ post–injection	55.5 ± 7.2 °C @10 min., ↑ 3.2 °C/min. [intra-]	×	75.7 ± 9.4 °C @10 min., ↑ 5.3 °C/min. [intra-]
				57.3 ± 49.3 cm^3^ @ Day 12[post–] <87.7 ± 15.8 cm^3^ @ Day 1[pre–]		163.4 ± 42.5 cm^3^@ Day 12[post–] ~132.9 ± 12.0 cm^3^ @ Day 1[pre–]
				Particle penetration into tissue & Uniform damages to tissue		Drops surrounding boundaries & Damages to only boundaries
			Only injection	236.9 ± 37.9 cm^3^@ Day 12[post–] >123.6 ± 18.5 cm^3^ @ Day 1[pre–]	×	454.5 ± 72.7 cm^3^@ Day 12[post–] >>98.7 ± 10.9 cm^3^ @ Day 1[pre–]
		CTL & Exp. Group	No injection	632.1 ± 82.2 cm^3^ @ Day 12 >> 110.7 ± 13.3 cm^3^ @ Day 1 (whether the post–AC magnetic field or not)		

## Data Availability

The data presented in this study are available upon request from the corresponding author. The data are not publicly available due to funders, and so cannot be made freely available.

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
