# Peer review of "Synthesised Conductive/Magnetic Composite Particles for Magnetic Ablations of Tumours"

_micromachines, 2022, doi:10.3390/mi13101605_

Round 1

Reviewer 1 Report

The manuscript describes the use of a Ga/Iron oxide composite for treatment of tumor. The work has some weaknesses that compromise its quality and require review to achieve quality in order to be published. The English needs a thorough revision. There are many confusing passages that prevent understanding of the subject and several spelling errors.

The experimental design needs to be better described. I suggest a figure containing the outline of the experimental groups and the controls. It is also necessary to rewrite the results pointing out how the treated groups behaved compared to the controls.

The figures need to be completely redone. It is necessary to include the error bars (standard deviation) in figures 2 and 3 (a, b). Without this it is not possible to evaluate the results. It is mandatory to clearly present the magnification of the photomicrograph obtained in SEM (figure 1a) and present an image without distortion in the AFM. Without these modifications, it is not possible to assess what the authors present in the text.

Regarding the images of the animals, the authors need to present them in a size that allows a clear visualization of the effect they claim to have obtained in the treatments applied. Considering that these images are the key point of the article, I strongly suggest that the authors provide images that allow the unequivocal visualization of the results obtained.

Finally, the discussion presented by the authors was limited to the restatement of the results, with excerpts containing copy and paste, and therefore did not add anything to the quality of the article. I strongly suggest a revision of this section to meet the quality of publication in the Micromachines journal

Reviewer 2 Report

The manuscript “The Synthesized Conductive/Magnetic Composite Particles for Tumor Magnetic Ablations” by Chiang-Wen Lee et al. proposed the synthesized particles with dual mechanisms of the induced heat by AC magnetic fields, and the tumor treatment by magnetic ablation, which claimed to be superior to magnetic particle hyperthermia in heating tissue.

The manuscript fits well into the keywords of this MDPI journal, and it has interesting and reasonably convincing parts. However, there are some clear shortcomings to correct prior to possible publication.

Authors describe iron oxide magnetic particles with clear focus on thermal ablation or hyperthermia. There are both well-known basic reviews and very recent publications on this subject – they should be used to improve the quality of the Introduction. In particular, the concept of the usage of the nanoparticles would be better to extend by the clear indication that the particles themselves can not be used for biological applications, the water-based suspensions are used in biomedicine. Authors may wish to analyze following works or select others (Moroz et al. J. Surg. Oncol. 2001, 77, 259–269.Coisson et al. J. Magn. Magn. Mater. 2016415, 2–7; Kurlyandskaya et al. Sensors 2017, 17(11), 2605; etc.). One of the missed concepts is the possibility to use such composites as ferrogels – being to some extent closer to proposed in this work composites in comparison with the suspensions.

In addition, there is a very important condition for the success of any biomedical application – the size of the unique batch (Grossman et al. Nanotechnology in cancer medicine. Phys. Today 201265, 38–42; Safronov et al. AIP Adv. 2013, 3, 52135.). The data on the batch size for the proposed material would be an advantage.

The nanoparticle size distribution must be given (obtained by SEM, TEM or DSL) and XRD + titration data would are necessary for the confirmation of the iron oxide particular composition.

VSM is well-known technique and there is no need to describe the principle of the device. Primary magnetization curves (they are present in the figure 2) can be used for the estimation of the spherical shape of the nanoparticles ( see description of the method in Ramachandran, et al. Solid State Commun. 199596, 127–131) – it is important for the evaluation of the average size of the particles by XRD.

The estimation of the experimental errors must be included (see Figure 3 as an example). 

Round 2

Reviewer 2 Report

Authors made special effort and significantly improved the quality of the manuscript answering all questions at sufficient level. There are some misprints and wrong writings of the names in the references but they can be corrected during proof-reading.